# Herpes Simplex 2 Virus Depletes Cells of DEAD-Box Helicase 3 Protein by Packaging It into Virions

**DOI:** 10.3390/v17081124

**Published:** 2025-08-15

**Authors:** Carmen Rita Piazza, Giulia Lottini, Paola Quaranta, Paola Perrera, Fabio Filippini, Michele Lai, Cristina Di Primio, Giulia Freer, Mauro Pistello

**Affiliations:** 1Retrovirus Center, Department of Translational Medicine and New Technologies in Medicine and Surgery, University of Pisa, 56127 Pisa, Italy; carmen.piazza@med.unipi.it (C.R.P.); lottini.g@gmail.com (G.L.); paola.quaranta@unipi.it (P.Q.); paola.perrera@gmail.com (P.P.); f.filippini1@studenti.unipi.it (F.F.); michele.lai@unipi.it (M.L.); mauro.pistello@unipi.it (M.P.); 2Virology Unit, Pisa University Hospital, 56127 Pisa, Italy; 3Institute of Neuroscience, National Research Council of Italy, 56127 Pisa, Italy; cristina.diprimio@in.cnr.it

**Keywords:** DEAD-box protein-3, herpes simplex virus 2, host factors for viral replication

## Abstract

Human DEAD-box helicase 3 (DDX3) is a multifunctional RNA helicase implicated in mRNA unwinding and the regulation of gene expression. While DDX3 has been extensively studied in the context of RNA virus replication, its role in DNA virus replication remains less understood. In this study, we explore the involvement of DDX3 in the life cycle of Herpes Simplex Virus type 2 (HSV-2), a double-stranded DNA virus. Silencing of DDX3 expression with siRNA significantly impaired HSV-2 replication, indicating that DDX3 supports viral propagation. Unexpectedly, HSV-2 infection led to a marked reduction in cellular DDX3 protein levels during in vitro replication in human cells, particularly at 24 h post-infection, corresponding to the peak of viral production. Notably, this decrease was not accompanied by a reduction in DDX3 mRNA levels, nor was it prevented by proteasome inhibition, suggesting an alternative mechanism of DDX3 depletion. Further analysis revealed substantial amounts of DDX3 protein within HSV-2 virions, supporting the hypothesis that DDX3 is packaged into viral particles during replication. We propose that HSV-2 exploits host DDX3 by incorporating it into progeny virions to facilitate early stages of infection in newly infected cells. However, no evidence linking DDX3 to the assembly process of HSV-2 particles was found. These findings expand the known functional repertoire of DDX3 and highlight its potential as a host factor co-opted by DNA viruses, suggesting a broader relevance in antiviral strategies.

## 1. Introduction

The human X-linked ATPase/RNA helicase DEAD-box polypeptide 3 (DDX3) is a member of the Asp-Glu-Ala-Asp (DEAD)-box protein family characterized by its ATPase domain and RNA-unwinding helicase activity [1,2]. DDX3 shuttles between the nucleus and the cytoplasm and plays a key role in various aspects of RNA metabolism processes, as well as in numerous physiological and pathological cellular functions [3]. Notably, in recent years, DDX3 has been shown to be exploited by several positive-sense single-stranded RNA viruses, such as the hepatitis C virus [3], Japanese encephalitis virus [4], Dengue virus [5], West Nile virus [6], enterovirus 71 [7], and human immunodeficiency virus type 1 [8]. Furthermore, we have demonstrated the efficacy of novel DDX3 inhibitors against Coxsackie B5 virus (CVB5), while observing no significant effect against negative-sense single-stranded RNA viruses, such as the measles virus and the vesicular stomatitis virus [9].

While numerous reports highlight the critical role of DDX3 in RNA virus infections, its involvement in the life cycle of DNA viruses remains relatively understudied [10]. Emerging evidence suggests that DNA viruses, such as herpesviruses, are influenced by DDX3 activity. For instance, DDX3 regulates the expression of viral genes during herpes simplex type 1 (HSV-1) infection [11,12] and has also been shown to promote cytomegalovirus replication [13,14]. Additionally, the Hepatitis B virus, another DNA virus, interacts with DDX3 to evade the host immune response, indicating that DDX3 may be exploited by DNA viruses to hamper cellular defenses [15].

More recently, HSV-1 was reported to redirect DDX3 to the nuclear membrane, where it is thought to play a role in modulating capsid budding from the nucleus and facilitating the release of progeny viral particles [12]. In the present study, a decrease in DDX3 levels in cells at late stages of HSV-2 infection was also found. This finding appears to contrast with the fact that HSV-2, like HSV-1, depends on DDX3 for efficient replication. Therefore, the aim of this study was to investigate the mechanisms responsible for DDX3 downregulation during HSV-2 infection and to determine the functional consequences of DDX3 diminished activity on viral replication. Our results suggest that DDX3 is incorporated into assembling virions and exported from the cell, thereby reducing intracellular DDX3 levels, whereas there was no evidence of a role of DDX3 in HSV-2 assembly.

## 2. Methods

### 2.1. Cells and Viruses

Human lung adenocarcinoma cells (A549) were cultured in high-glucose DMEM medium (Gibco; Thermo Fisher Scientific, Waltham, MA, USA) supplemented with 7% heat-inactivated fetal bovine serum (FBS) (Thermo Fisher Scientific, Waltham, MA, USA) and 2 mM of L-glutamine at 37 °C in the presence of 5% CO_2_.

The HSV-2 neurovirulent strain MS (ATCC VR-540) was used throughout. It was propagated on Vero E6 cells. Briefly, cells were infected at approximately 90% confluence with 0.001 MOI of HSV-2 in medium without FBS. Cells were incubated at 37 °C and 5% CO_2_ for 1 h, with flasks shaken every 15 min. At the end of the incubation, culture medium supplemented with 2% FBS was added. After 48 h post-infection (p.i.), supernatant was collected, centrifuged at 648× *g* for 10 min, aliquoted, and stored at −80 °C. Viral titer was calculated by plaque assay and expressed as plaque-forming units (PFU)/mL. CVB5 was grown on HeLa cells, and Chikungunya (CHIKV) and Zika viruses (ZIKV) were propagated on Huh-7.

### 2.2. Plaque Assays

Plaque assays were performed as described [16]. Vero E6 cells were seeded in 24-well plates and infected with HSV-2 1:5 serial dilutions for 1 h at 37 °C and 5% CO_2_. After infection, plates were overlaid with 0.75% Methyl Cellulose (Sigma-Aldrich, St. Louis, MO, USA) in culture medium supplemented with 2% FBS. Cells were incubated for 48 h, then they were fixed in paraformaldehyde overnight (O/N) and stained with crystal violet (Sigma-Aldrich).

### 2.3. mRNA Extraction from Cell Lysates

To evaluate DDX3 gene expression, A549 cells (2 × 10^5^) were seeded for 24 h. The next day, the cell monolayer was infected by HSV-2, MOI 0.05, CVB5 MOI 0.0005, CHIKV MOI 500 or ZIKV MOI 0.5 in DMEM without FBS for 1 h, then fresh medium with 2% FBS was added. At 0, 3, and 6 h p.i. for HSV-2 only, and at 16 and 24 h p.i. for all the viruses mentioned, cells were collected, and mRNA extraction was performed using TRIzol reagent (Invitrogen, Waltham, MA, USA) following the manufacturer’s instructions.

### 2.4. Real-Time PCR

For qRT-PCR on DDX3 mRNA, the reaction was set up by QuantiNova SYBR Green RT-PCR Kit with the following protocol: 10 µL 2× Quantinova master mix, 0.2 µL of QN RT-MIX, 0.5 µM primers (F: 5′-ACTATGCCTCCAAAGGGTGTCC-3′; R:5′-AGAGCCAACTCTTCCTACAGCC-3′), and 4 µL of extracted RNA in a total volume of 20 µL. The cycling conditions were 50 °C for 10 min, 95 °C for 2 min, 95 °C for 5 s, 60 °C for 10 s, 39 cycles of 95 °C for 5 s and 60 °C for 10 s, then a melt curve step from 65 °C to 95 °C with an increment of 0.5 °C for 20 s. Each sample was compared with its GAPDH RNA using the same conditions and primers: F: 5′- GTCTCCTCTGACTTCAACAGCG-3′; R: 5′- ACCACCCTGTTGCTGTAGCCAA-3′.

### 2.5. Silencing of DDX3 and Infection

SiRNAs against human DDX3X ON-TARGET plus Smartpool (Dharmacon, Lafayette, CO, USA) were used [11]. For transfection, Lipofectamine RNAi Max (Invitrogen, Thermo Fisher Scientific, Waltham, MA, USA) was mixed with siRNA, 0.2 µM, according to the manufacturer’s instructions. The solutions were subsequently combined with culture medium on A549 cells at a density of 3 × 10^5^ cells. In the negative control group, cell density was 2 × 10^5^ and did not undergo any treatment. In the transfection control groups, cells were transfected with a control siRNA (ON-TARGETplus Control Non-Targeting Pool) (Dharmacon, Lafayette, CO, USA). After 72 h, the medium was removed, and the cells were washed with PBS and infected with HSV-2, MOI 0.05. As a control virus, CVB5 or ZIKV was used, and the same protocol was applied, infecting with MOI 0.0005 and 0.5, respectively. Each sample had its own uninfected control. The cell lysates were prepared for protein extraction in RIPA Buffer at 16 and 24 h p.i.

### 2.6. Immunofluorescence of Infection Kinetics

A549 cells were infected with HSV-2, MOI 0.05 and samples were then harvested at 3, 6, 9, 12 and 24 h p.i. and analyzed by confocal microscopy as described below.

### 2.7. Proteasome Inhibition

A549 cells were treated with MG132, 10 and 1 µM, 60 min prior to infection. After washing with PBS, cells were infected with HSV-2, MOI 0.05 for 24 h, followed by protein extraction for Western blot (WB) analysis.

### 2.8. Ultracentrifugation of Supernatants

Supernatants from A549 cells infected with HSV-2 containing mature HSV-2 particles were purified by ultracentrifugation. Briefly, A549 cells were infected with HSV-2 at MOI 1 for 16 h. Extracellular medium was collected and centrifuged at 652× *g* for 10 min. Supernatants were then ultracentrifuged on a 20% sucrose cushion at 111,000× *g* for 2 h at 4 °C.

### 2.9. Western Blot

WB was performed as previously described [17]. Cells were washed twice with PBS and lysed in RIPA buffer supplemented with protease inhibitor cocktail (Sigma-Aldrich) and phosphatase inhibitor cocktail (Thermo Scientific). Following a 2 h incubation at 4 °C, lysates were centrifuged for 10 min at 4 °C at 13,000× *g*. Supernatants were mixed with 5× Laemmli sample loading buffer and heated at 70 °C for 10 min, then loaded onto 10% SDS-PAGE gels and transferred onto nitrocellulose membranes. Membranes were blocked with 5% BSA or skimmed milk in PBS-Tween 0.1% at room temperature (RT) for 1 h, then incubated with primary antibodies at 4 °C O/N. After washing, membranes were incubated with HPRO-conjugated anti-mouse IgG (1:20,000, A9044- Sigma-Aldrich, St. Louis, MO, USA) or anti-rabbit IgG (1:20,000, A0545- Sigma- Aldrich, St. Louis, MO, USA) and anti-goat IgG (1:2000, sc-2354-Santa Cruz, Dallas, TX, USA) at RT for 1 h. Protein bands were visualized using enhanced chemiluminescence reagents (Pierce ECL substrate, Thermo Fisher Scientific, Waltham, MA, USA) and acquired by ChemiDoc™ Imaging Systems (Bio-rad, Hercules, CA, USA). The primary antibodies used were as follows: mouse anti-ICP5 HSV (1:200, sc-56989-Santa Cruz, Dallas, TX, USA), rabbit anti-DDX3 (1:1000, D19B4- Cell signaling, Danvers, MA, USA), rabbit anti-GAPDH (1:2000, MA5-15738- Invitrogen, Thermo Fisher Scientific, Waltham, MA, USA), polyclonal human serum against CVB5 (1:200), rabbit anti-C ZIKV (1:1000, GTX133317- Genetex, Irvine, CA, USA), rabbit anti-E1 CHIKV (1:1000, GTX135187- Genetex, Irvine, CA, USA), rabbit anti-NS5 ZIKV(1:1000, GTX133328- Genetex, Irvine, CA, USA), mouse anti-p53 (1:2000, P6874- Merck, Darmstadt, Germany), rabbit anti-lamin A/C (1:1000, GTX101127- Genetex, Irvine, CA, USA), and goat anti-human HSP70 (1:1000, sc-1060-Santa Cruz, Dallas, TX, USA).

### 2.10. Confocal Imaging

Sample preparation was performed as described [17]. Briefly, A549 cells were infected with HSV-2 MOI 0.05 and fixed in 4% paraformaldehyde were washed three times in PBS, 0.1% Triton X-100 and permeabilized in PBS, 0.5% Triton X-100 for 10 min. Samples were then blocked for 1 h at RT in blocking buffer (PBS supplemented with 3% BSA, 0.3% Triton X-100, 0.1% goat serum and 1% human serum), then incubated O/N at 4 °C with the appropriate primary antibody diluted in antibody dilution buffer (PBS, 3% BSA, 0.2% Triton X-100, 0.1% goat serum and 1% human serum).

The next day, cells were washed in PBS and incubated with secondary antibody in antibody dilution buffer for 1 h at RT. Finally, cells were washed and images were acquired by Operetta CLS High-Content Imaging Device (Revvity, Waltham, MA, USA). To investigate the intensity of DDX3 fluorescence, the following building blocks were used: Find Nuclei > Find Citoplasm > Calculate intensity properties (DDX3-AlexaFluor647).

The following antibodies were used: rabbit monoclonal antibody anti-DDX3 (1:100, D19B4-Cell Signaling, Danvers, MA, USA); mouse monoclonal antibody anti-gD HSV (1:100, G610D- Invitrogen, Thermo Fisher Scientific, Waltham, MA, USA); mouse monoclonal antibody anti-ICP5 HSV (1:50, sc-56989-Santa Cruz, Dallas, TX, USA); mouse monoclonal antibody anti-ICP27 HSV (1:100, sc-69806-Santa Cruz, Dallas, TX, USA); mouse monoclonal antibody anti-VP16 HSV (1:100, sc-7545-Santa Cruz, Dallas, TX, USA); goat anti-rabbit IgG Alexa647-labeled antibody (1:1000, A-21245-Invitrogen, Thermo Fisher Scientific, Waltham, MA, USA); goat anti-mouse IgG Alexa488-labeled antibody (1:1000, A-11029- Invitrogen, Thermo Fisher Scientific, Waltham, MA, USA); and rabbit anti-goat IgG Alexa647-labeled antibody (1:1000, A-21446- Invitrogen, Thermo Fisher Scientific, Waltham, MA, USA). DAPI (1:5000, 28718-90-3- Sigma Aldrich, St. Louis, MO, USA) was added to stain nuclei.

### 2.11. Statistical Analysis

Data were analyzed with GraphPad Prism software, version 8 (GraphPad Software, San Diego, CA, USA) and numerical values expressed as mean ± standard deviation (SD) of the mean. *p* values < 0.05 were considered statistically significant.

## 3. Results

### 3.1. DDX3 Silencing Slows Down HSV-2 Replication

In previous work, inhibition of DDX3 activity in HSV-2-infected cells was observed to result in slower viral replication (unpublished observation). This finding was not straightforward to interpret, given that HSV-1 has been reported to exploit DDX3 activity during replication [11]. For this reason, one would expect the virus to enhance, rather than downregulate, DDX3 activity. Therefore, we sought to confirm and further investigate this observation.

First, we used siRNA to silence DDX3 (siDDX3) mRNA in human adenocarcinoma A549 cells for 72 h, using a scrambled siRNA (siCTRL) as a mock control. Cells were then infected with HSV-2 for 16 and 24 h. CVB5, which is known to rely on DDX3 for efficient replication [9], and ZIKV were used as control viruses. The effect of silencing DDX3 was assessed by measuring viral protein levels in cell lysates via WB analysis at 16 and 24 h p.i. As shown in Figure 1A, DDX3 protein levels were significantly reduced in siDDX3-treated A549 cells but remained clearly detectable in siCTRL-treated cells. The amount of HSV-2 major capsid protein ICP5 was markedly decreased in siDDX3-treated cells at both time points, while siCTRL-treated cells showed ICP5 levels similar to untreated controls. Figure 1B also shows the effect of DDX3 silencing on CVB5 replication; in these experiments, CVB5 coat protein levels decreased in siDDX3-treated cells, although to a lesser extent than in HSV-2-infected cells. Similar results were also observed with ZIKV as a control virus.

To further confirm that HSV-2 replication was impaired following DDX3 silencing, we measured HSV-2 titers in supernatants from siDDX3- or siCTRL-treated cells at 16 and 24 h p.i. by plaque assay. As expected, HSV-2 titers in supernatants from siDDX3-treated cells were approximately 2 logs lower than in siCTRL-treated or untreated controls (Figure 1D). In addition, we analyzed syncytium formation in HSV-2-infected siDDX3-treated cells by confocal microscopy (Figure 1E). Consistent with other results, siDDX3-treated cells showed a markedly reduced number of HSV-2 syncytia, whereas numerous syncytia were observed both in siCTRL-treated and untreated cells. Collectively, these data suggest that DDX3 plays an important role in HSV-2 replication.

### 3.2. HSV-2 Infection Leads to a Reduction in DDX3 Protein Levels

In many cases, viruses manipulate host protein expression to facilitate their own replication, leading to the upregulation of specific host proteins that support various stages of the viral life cycle [4,18]. To investigate whether DDX3 expression changed during HSV-2 replication in susceptible cells, A549 were infected with HSV-2, alongside other viruses, such as CVB5, CHIKV and ZIKV, as controls. DDX3 protein levels were measured by WB at 16 and 24 h p.i. Viral infection was confirmed by probing for proteins specific to each virus, while GAPDH served as a loading control (Figure 2A). As expected, an increase in the amount of HSV-2 ICP5 was observed between 16 and 24 h p.i. Similarly, increases in NS5, E1, and total capsid were found for ZIKV, CHIKV, and CVB5, respectively. Surprisingly, however, DDX3 protein levels were found to be significantly reduced in HSV-2-infected cells, as early as 16 h p.i and remained decreased at 24 h. This reduction was not attributable to general cellular damage or lysis caused by HSV-2 infection, as GAPDH levels remained stable. Furthermore, under the described experimental conditions, cytopathic effects in A549 cells at 16 or even 24 h p.i. were minimal (see Figure 2).

Quantitative analysis indicated that HSV-2 infection caused up to a 23% reduction in DDX3 levels after normalization to GAPDH. In contrast, no significant changes in DDX3 levels were observed in cells infected with CVB5 or the other control viruses (Figure 2A,B).

To assess the kinetics of DDX3 reduction during HSV-2 infection, we performed a confocal microscopy time-course analysis of cells stained for HSV-2 ICP27 (an immediate early protein), VP16 (a late protein), and DDX3 at 3, 6, 9, 12, and 24 h p.i. (Figure 2C,D). A clear decrease in DDX3 was observed only at later stages (around 24 h p.i.), coinciding with strong VP16 expression. These findings suggest that DDX3 reduction occurs during the late phase of infection, when virion assembly and release are underway.

### 3.3. DDX3 Protein Does Not Undergo Proteasome Degradation or Transcriptional Shutoff During HSV-2 Infection

To understand why HSV-2 infection, but not infection by other control viruses, led to significant reduction in DDX3 protein levels, a role for proteasomal degradation and/or viral shutoff were hypothesized. The proteasome plays a relevant role in protein turnover by degrading ubiquitinated proteins, such as p53, which is a well-known proteasome substrate [19]. Thus, we first assessed whether proteasomal degradation could account for the decrease in DDX3: A549 cells were treated with MG132, a proteasome inhibitor, 1 h prior to HSV-2 infection. Cells were lysed at 24 h p.i. and analyzed by WB. Proteasomal activity was monitored by quantifying p53, indicating successful MG132 treatment (Figure 3A). While p53 was spared from degradation due to MG132, DDX3 protein levels remained unchanged in MG132-treated cells compared to untreated controls (Figure 3B), suggesting that the reduction of DDX3 during HSV-2 infection is not mediated by proteasomal degradation.

We next explored whether viral-induced transcriptional shutoff could account for the decrease in DDX3 protein. HSV-2 encodes several proteins known to disrupt host gene expression, including the virion host shutoff (VHS) protein, encoded by the *UL41* gene. This protein acts as a ribonuclease, degrading host mRNAs in the cytoplasm [20]. Additionally, ICP27 inhibits host RNA splicing, thereby hindering the export of mature mRNAs to the cytoplasm and leading to shutoff of protein synthesis [21]. To address this issue, we designed DDX3-specific primers spanning different exons flanking the same intron and used them to quantify mRNA encoding DDX3 in infected and uninfected cells. Total RNA was extracted from HSV-2–infected and uninfected A549 cells at 0, 3, 6, 16 and 24 h p.i., and DDX3 mRNA levels were measured by qRT-PCR. After normalization with GAPDH, mRNA levels were compared between infected and uninfected cells for each time point, and, as shown in Figure 3C, mRNA levels in HSV-2-infected cells were comparable to those in uninfected controls at 0, 3, 6 h p.i. At 16 and 24 h p.i., mRNA levels were found to be even increased. These results indicate that the decrease in DDX3 protein was not due to reduced transcription of the corresponding gene. As for control viruses, DDX3 mRNA levels were measured by qRT-PCR in uninfected A549 or infected with CVB5, CHIKV or ZIKV at 16 and 24 h p.i. After normalization with GAPDH, mRNA levels were compared between infected and uninfected cells for each time point and mRNA levels in infected cells were comparable to those in uninfected controls (Figure 3D).

These results collectively suggest that the reduction in DDX3 protein levels during HSV-2 infection is not the result of proteasomal degradation or host transcriptional shutoff.

### 3.4. DDX3 Protein Localization in HSV-2 Virions and in HSV-2-Infected A549 Cells

Once a role for the proteasome and transcriptional shutoff had been ruled out, we investigated whether DDX3 could be found in extracellular virions. DDX3 has previously been detected in herpesvirus extracellular virions by several authors [12,22,23]. To examine extracellular virions for the presence of DDX3 in this work, we purified virions from the supernatant of HSV-2-infected cells at 16 h p.i. on a sucrose cushion and analyzed them by WB. In addition to DDX3 and ICP5, we assessed the presence of Lamin A/C (a nuclear marker), HSP70 (a marker for exosomes [24]) and GAPDH (a cytoplasmic marker). As shown in Figure 4A, DDX3 can be found in extracellular virions, whereas the absence of reactivity to HSP70 demonstrates that exosomal contamination could be ruled out.

Recent work has described DDX3 as instrumental in the budding of HSV-1 on the nuclear membrane, where it was shown to colocalize with HSV-1 capsid protein at early times of infection [12]. To assess whether DDX3 could also be found on the nuclear membrane of HSV-2-infected cells, we performed confocal microscopy at 8 and 16 h p.i. As a caveat, we considered that the anti-DDX3 antibody is raised in rabbit. To avoid the property of HSV gE/gI complex to bind the Fc region of rabbit and human (but not goat or mouse) antibodies [25,26], fixed cells were first incubated in 1% human serum to block nonspecific Fc receptor binding at sites where HSV-2 envelope proteins may be present. Infected cells at 16 h p.i. were stained with an irrelevant fluorescently labeled rabbit antibody (rabbit anti-goat IgG). This control confirmed that nonspecific staining persisted despite blocking with human serum (Figure 4B). Cells were then stained for the immediate early ICP27 and late capsid ICP5 at 8 h p.i., and for ICP5 or gD at 16 h p.i. As shown in Figure 4C, at HSV-2 capsid assembly sites, DDX3 staining was not more prominent than anywhere else (middle fourth/fifth panel from the left). These observations suggest that DDX3 is incorporated into virions at a later stage. Consequently, we conclude that DDX3 associates with virions after nuclear egress, likely during tegumentation, which occurs in the cytoplasm [23,27].

## 4. Discussion

DDX3 is a human ATP-dependent RNA helicase with multiple functions. It participates in various cellular processes, including cell cycle regulation and protein translation, where it associates with translation initiation complexes [28,29]. Notably, DDX3 plays a critical role in the translation of mRNAs that possess long or structurally complex 5′ untranslated regions, likely by unwinding secondary structures during translation [30]. Several studies have shown that DDX3 can function either as a host restriction factor or as a pro-viral component. For RNA viruses such as the hepatitis C virus, flaviviruses, and enteroviruses, DDX3 facilitates replication by enhancing translation and promoting viral RNA export [2,9,31,32]. In this study, we investigated the role of DDX3 during HSV-2 infection, prompted by the observation that DDX3 protein levels change during infection. Given that HSV-2 is a DNA virus, the function of DDX3, an RNA helicase, in this context remained unclear.

First, the effect of DDX3 on HSV-2 infection was investigated by silencing DDX3 expression. It was confirmed that, in the absence of DDX3, viral replication was considerably slower, with a reduced accumulation of viral protein in HSV-2-infected cells. Additionally, the number of syncytia was markedly lower in HSV-2-infected cells lacking DDX3, and HSV-2 yields in supernatants from these cells were between 50 and 150 times lower. These results are consistent with those of Khadivjam et al. [11], who showed that another herpesvirus, HSV-1, also requires DDX3 for efficient replication in various cell types. Their findings can be extended to HSV-2 in A549 human cells, where DDX3 appears essential for efficient viral replication. Given the high degree of homology between HSV-1 and HSV-2 [33], exceeding 90% at the amino acid level, this result is not unexpected [34].

Given its important role in HSV-2 replication, one might expect DDX3 to be up-regulated during HSV-2 infection, as observed for other host factors hijacked by viruses [18]. However, in our experiments, DDX3 protein levels were much lower in cells during late phases of HSV-2 infection. In contrast, infection with CHIKV, ZIKV or CVB5 did not affect DDX3 intracellular levels. All these viruses are known to shutoff host protein expression; in herpesvirus infection, viral proteins, such as VHS and ICP27, play key roles in this process through different mechanisms that ultimately reduce host mRNA levels. To determine whether shutoff affects DDX3 mRNA, DDX3 transcription in HSV-2-infected cells was assessed by qRT-PCR. While DDX3 mRNA levels remained unchanged at early stages of infection (0, 3, 6 h p.i.), at 16 and 24 h p.i., there appeared to be an increase in DDX3 expression, indicating that the reduction in cellular DDX3 protein during HSV-2 infection was not due to transcriptional shutoff. Proteasomal degradation of DDX3 was also ruled out by experiments using MG132, a proteasome inhibitor. While MG132 successfully prevented p53 degradation, as expected, it did not lead to an increase in DDX3 protein levels. Thus, proteasomal degradation does not seem to target this protein in HSV-2-infected cells.

DDX3 has been reported to be packaged into HSV-1 and other DNA virus virions [11,12,15]. In the present study, WB analysis of extracellular HSV-2 virions revealed that DDX3 is also incorporated in significant amounts. This finding adds to the growing body of evidence that herpesviruses hijack host cellular machinery during virion assembly, potentially incorporating host-derived factors into the viral particle. To our knowledge, the precise localization of DDX3 within the virion has not been elucidated in the literature. Our results suggest that DDX3 is acquired at a stage following primary nuclear budding, as no colocalization with capsid proteins or surface glycoproteins was detected by confocal microscopy. Given that tegumentation takes place in the cytoplasm, it is plausible that this is the stage at which DDX3 becomes incorporated into the virion [27].

Khadivjam et al. explained the presence of DDX3 in virions by suggesting that DDX3 might assist in virion assembly; accordingly, they visualized DDX3 aggregates at the nuclear periphery together with the capsid protein. In our experiments, however, we did not detect DDX3 at the nuclear rim during HSV-2 replication, although we did find DDX3 inside extracellular virions. Several factors might explain this discrepancy: Khadivjam, et al. studied HSV-1 in HeLa cells, whereas we focused on HSV-2 in A549 in the present work. Notably, HeLa cells have a female genetic background, whereas A549 cells are male. Lastly, human serum was added in our immunofluorescence experiments because rabbit antibodies, used to stain DDX3, bind to HSV-2-infected cells via a virally encoded Fc receptor [25]. Human serum blocks this reaction for the most part since human IgG, such as rabbit antibodies, bind to the HSV gE/gI complex.

We hypothesize that cells are depleted of DDX3 because HSV-2 packages significant amounts of this protein into its virions, thereby exporting it out of the cell. HSV tegument proteins are known to interact with and carry cell transcription factors, such as HCF-1 and Oct-1, and other host proteins [27]. These interactions likely occur in the cytoplasm [35] and are very important for early viral gene expression. We consistently saw a decrease in cellular DDX3 at the time of virion release, when cells were full of viral structural proteins, but not earlier, when replication is active but no virions had yet exited the infected cells. Furthermore, no shutoff affecting transcription of the DDX3 gene was observed, nor did we find evidence of proteasomal degradation of DDX3 upon HSV-2 infection. Shutoff and proteasomal activation are typically early events in alphaherpesvirinae replication [36].

The question of why HSV-2 packages DDX3 in virions was not directly addressed in this study; however, the most obvious hypothesis is that it could unwind RNA transported within virions or newly synthesized early viral mRNA [37]. Supporting this, RNAs have been found encapsidated in HSV virions [37]. Another possibility is that DDX3 packaging is an incidental consequence of its interaction with viral replication proteins, leading to its inadvertent incorporation. This seems unlikely, though, given the large amount of DDX3 that we and others have found in virions (see Figure 4A). In addition, no evidence of colocalization with HSV-2 replication sites was found in our experiments.

In conclusion, we hypothesize that the depletion of cellular DDX3 results from its packaging into mature virions, but we do not provide evidence supporting a direct role in viral assembly. These results reinforce the role of DDX3 as a host factor hijacked by HSV-2, similar to HSV-1. While largely consistent with previous studies, our data challenge the notion that DDX3 aids HSV assembly, at least in the case of HSV-2.

## Figures and Tables

**Figure 1 viruses-17-01124-f001:**
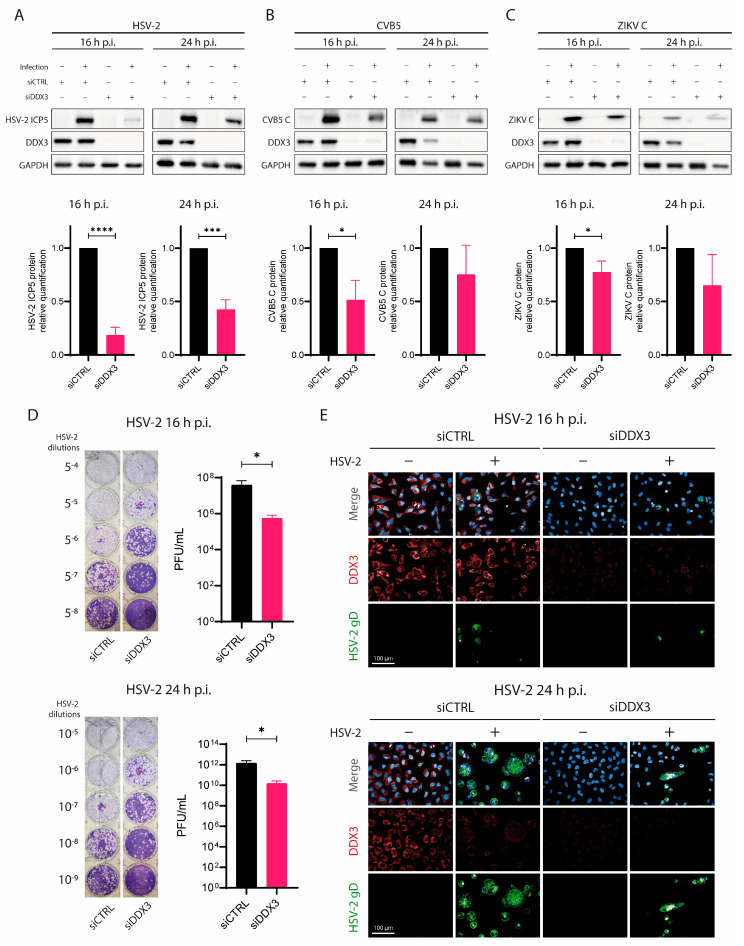
DDX3 silencing slows down HSV-2 replication. A549 cells were transfected with siDDX3, siCTRL for 72 h, then they were infected with (**A**) HSV-2, (**B**) CVB5 or (**C**) ZIKV. At 16 h and 24 h p.i., cells were lysed and analyzed for DDX3, GADPH, (**A**) HSV-2 viral ICP5 protein, (**B**) CVB5 coat protein, (**C**) or ZIKV C by WB. Lower graphs depict the fold change of each protein normalized to GADPH. Statistical analysis was performed by *t*-test; * *p* < 0.05, *** *p* < 0.001, **** *p* < 0.0001. (**D**) Supernatants from cells used in (**A**) were titrated in 24-well plates on Vero E6 cells for HSV-2 at 16 (upper panel) and 24 h p.i. (lower panel). The image shows PFUs visualized after crystal violet staining. PFU/mL in supernatants were plotted on the right of each plaque assay and statistical analysis was performed by *t*-test; * *p* < 0.05. (**E**) Confocal microscopy analysis of A549 transfected with siCTRL or siDDX3 for 72 h, then infected or not with HSV-2. Cells were fixed 16 h (upper panel) or 24 h (lower panel) p.i., permeabilized and stained with DAPI (blue), anti-HSV-2 gD (green) and anti-DDX3 (red) antibodies. Magnification 40×. Scale Bar: 100 µm. Data are shown as mean ± SD (N = 3).

**Figure 2 viruses-17-01124-f002:**
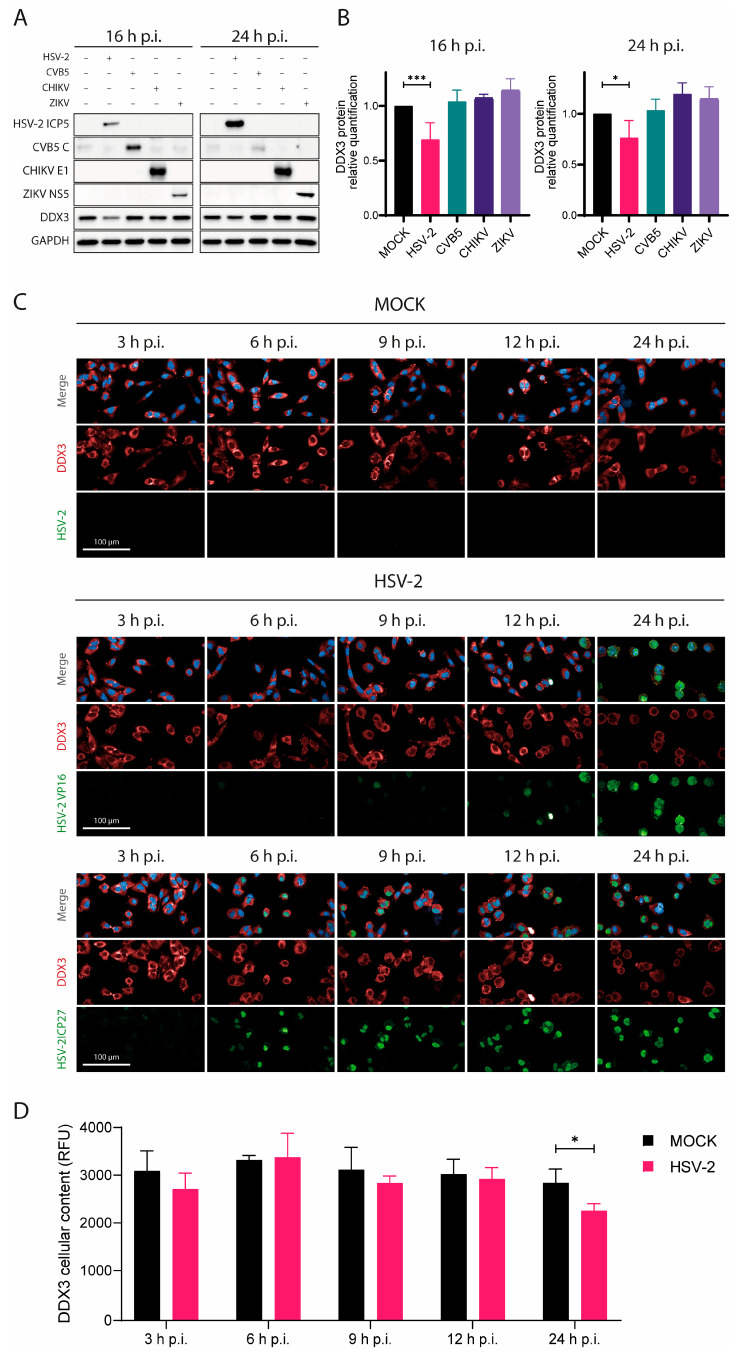
HSV-2 infection leads to progressive downregulation of the DDX3 protein. (**A**) A549 cells were infected with HSV-2, CVB5, CHKV or ZIKV for 16 and 24 h, then cells were lysed for WB analysis. Antibodies against either HSV-2 ICP5, CBV5 capsid, CHKV E1 or ZIKV NS5, DDX3 and GADPH were used. (**B**) DDX3 protein was quantified relative to its amount in untreated cells in each WB in (**A**) and statistical analysis was performed by one-way ANOVA post hoc Tukey’s test; * *p* < 0.05, *** *p* < 0.001. (**C**) A549 cells were infected with HSV-2 and fixed at 3, 6, 9, 12, and 24 h p.i. Cells were stained for DDX3 protein (red), VP16 (green, middle panels), ICP27 (green, bottom panels) and DAPI (blue). Magnification 40×. Scale Bar: 100 µm. (**D**) Analysis of the panels in (**C**) at 3, 6, 9, 12, and 24 h p.i. DDX3 protein was quantified based on its relative fluorescence (RFU) in untreated cells by two-way ANOVA post hoc Sidak’s test; * *p* < 0.05. Data are shown as mean ± SD (N = 4).

**Figure 3 viruses-17-01124-f003:**
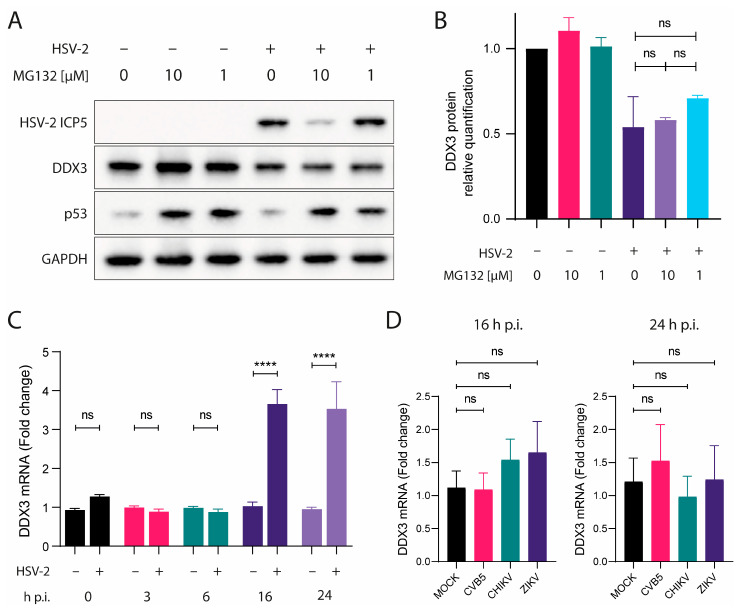
DDX3 does not undergo proteasome degradation or transcriptional shutoff during HSV-2 infection. (**A**) A549 cells were treated with MG132 for 60 min, then infected with HSV-2. Cells were lysed after 24 h p.i. and analyzed by WB with anti-ICP5, -DDX3, -p53 or -GAPDH antibodies. (**B**) DDX3 protein was quantified from data in (**A**) relative to its amount in untreated cells. One-way ANOVA post hoc Tukey’s test; ^ns^ *p* > 0.05. (**C**) DDX3-encoding mRNA in lysates of cells infected or not with HSV-2 at 0, 3, 6, 16 and 24 h p.i. or (**D**) CVB5, CHIKV and ZIKV at 16 and 24 h p.i. was quantified relative to its amount in uninfected cells. Statistical analysis was performed by one-way ANOVA post hoc Dunnett test; ^ns^ *p* > 0.05, **** *p* < 0.0001. Data are shown as mean ± SEM. N = 4.

**Figure 4 viruses-17-01124-f004:**
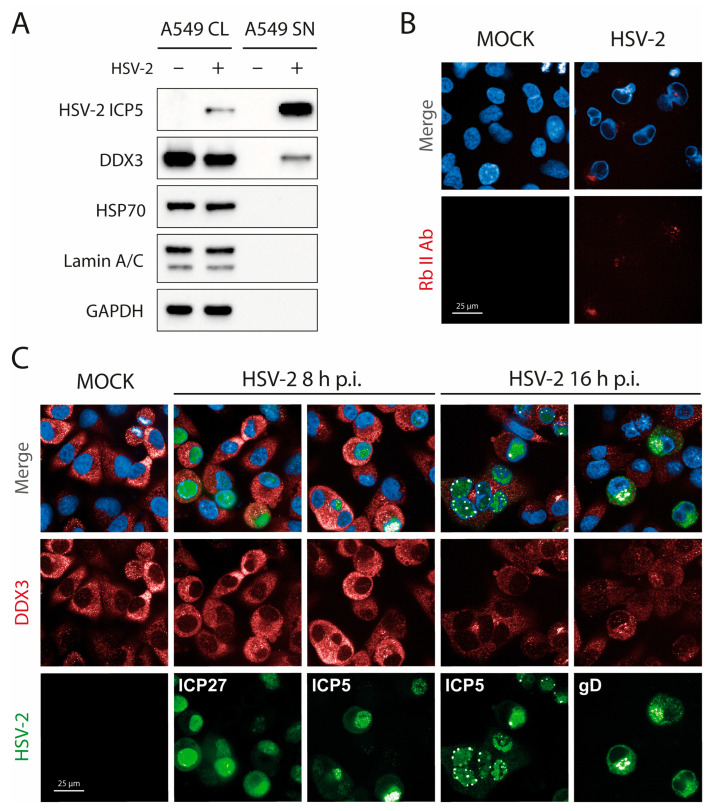
DDX3 can be found in extracellular HSV-2 virions but its localization does not change during HSV-2 infection. (**A**) HSV-2 virions were purified on a sucrose cushion and then analyzed by WB for ICP5, DDX3, HSP70 as an exosome marker, Lamin A/C as a cell nucleus marker and GAPDH as a cell cytoplasm control. (CL = Cell Lysate; SN = Supernatant) (N = 2). (**B**) A549 cells were infected with HSV-2 and 16 h p.i.; they were fixed, blocked with human serum, and stained with a nonspecific rabbit anti-goat-Alexa647 antibody. (**C**) A549 cells were infected with HSV-2 and 8 or 16 h p.i.; they were fixed and stained with antibodies against DDX3 (red), late ICP5 (green), and early ICP27 (green, 8 h p.i.) or late gD (green, 16 h p.i.) and DAPI (blue). Immunofluorescence was performed by adding human serum (1%) in blocking and antibody solutions. Magnification 63×. Scale Bar: 25 µm.

## Data Availability

The original contributions presented in this study are included in the article. Further inquiries can be directed to the corresponding author.

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
