# Peer review of "Herpes Simplex 2 Virus Depletes Cells of DEAD-Box Helicase 3 Protein by Packaging It into Virions"

_viruses, 2025, doi:10.3390/v17081124_

Round 1
Reviewer 1 Report
Comments and Suggestions for Authors
Please see the attachment

Section 3.4 is too short and lacks rationales.
Reviewer 2 Report
Comments and Suggestions for Authors
In this manuscript, the authors investigate the role of DEAD-box helicase 3 (DDX3) in herpes simplex virus type 2 (HSV-2) infection, which plays a role in the replication of multiple (+)sense RNA viruses. Antisense knockdown of DDX3 mRNA negatively impacted HSV gene expression, viral titers, and syncitia at 16 and 24 hours post-infection. First by Western blotting and then immunofluorescence, the authors noticed that DDX3 levels decreased at later times of infection despite its beneficial role in virus replication. This loss was not reduced by using MG132 to inhibit proteasome activity, and considering the loss at late times of infection, it was hypothesized that DDX3 is incorporated into viral particles. mRNA levels were found not to decrease in HSV infection, but were with chikungunya virus. DDX3 was also found not to grossly change localization with microscopy of HSV-2-infected cells, which differs from one report which used the related virus HSV-1. The conclusions are strong about the loss of DDX3 at late timepoints, and this is not due to proteasomal degradation, while there is DDX3 in the supernatant of infected cells. The discussion of rabbit antibodies binding to HSV Fc receptors is an important message to the field, this also cost some delays in my research until speaking with other experts about some odd results. Overall, the paper provides an interesting consideration for a factor that influences a broad range of virus families and an alluring hypothesis for a proviral factor being nearly redirected to viral particles. However, there are a few controls or adjustments that would greatly strengthen some of the conclusions as described below.
Major recommendations:
It would be helpful to purify virions away from extracellular bodies like exosomes to show that DDX3 is indeed packaged into virions rather than sent to other exit pathways in the cell. This could be accomplished by a ficoll or OptiPrep gradient, or show that the sucrose cushion is sufficient to separate virions from these particles. It would also be useful to show a lack of DDX3 from supernatant from uninfected cells spun over the sucrose gradient.
I believe it would also be helpful to show the ZIKV / CHIKV and DDX3 data as it is referenced three times in two sections of the document, and provide useful comparisons.
As the authors discuss with UL41, this nuclease will degrade translated mRNA-including GAPDH which is the calibration control for qPCR. Therefore, I wonder if the lack of change in DDX3 mRNA in HSV-2-infected cells is actually more of an equal rate of degradation as GAPDH (both shift an equal number of cycles). I would therefore propose to use rRNA instead as an internal calibration control, or perhaps provide the raw Ct values in the response or as an appendix table. Even if there is some shift, it does not harm the major conclusions of the paper, but at least provides better clarification of the ability of VHS/UL41 to prevent synthesis of new DDX3.
Minor text edits:
The first paragraph of each section lacks indentation
Italicize et al. throughout the document
Line 261: place a space in ".To"
Line 294: place a space in "andICP5"
Lines 388 and 391- place a space after the period in the section header
Round 2
Reviewer 1 Report
Comments and Suggestions for Authors
Most of the concerns are addressed.